# Emerging Southeast Asian PfCRT mutations confer *Plasmodium falciparum* resistance to the first-line antimalarial piperaquine

Leila S. Ross[1], Satish K. Dhingra[1], Sachel Mok [1], Tomas Yeo [1], Kathryn J. Wicht[1], Krittikorn Kümpornsin [1,8], Shannon Takala-Harrison[2], Benoit Witkowski[3], Rick M. Fairhurst[4,9], Frederic Ariey[5], Didier Menard[3,6] & David A. Fidock[1,7]

The widely used antimalarial combination therapy dihydroartemisinin + piperaquine (DHA + PPQ) has failed in Cambodia. Here, we perform a genomic analysis that reveals a rapid increase in the prevalence of novel mutations in the *Plasmodium falciparum* chloroquine resistance transporter PfCRT following DHA + PPQ implementation. These mutations occur in parasites harboring the K13 C580Y artemisinin resistance marker. By introducing PfCRT mutations into sensitive Dd2 parasites or removing them from resistant Cambodian isolates, we show that the H97Y, F145I, M343L, or G353V mutations each confer resistance to PPQ, albeit with fitness costs for all but M343L. These mutations sensitize Dd2 parasites to chloroquine, amodiaquine, and quinine. In Dd2 parasites, multicopy *plasmepsin 2*, a candidate molecular marker, is not necessary for PPQ resistance. Distended digestive vacuoles were observed in *pfcrt*-edited Dd2 parasites but not in Cambodian isolates. Our findings provide compelling evidence that emerging mutations in PfCRT can serve as a molecular marker and mediator of PPQ resistance.

[1] Department of Microbiology and Immunology, Columbia University Irving Medical Center, 1502 Hammer Health Sciences Center, 701 W. 168th St., New York, NY 10032, USA. [2] Center for Vaccine Development and Global Health, University of Maryland School of Medicine, 685 W. Baltimore Street, Baltimore, MD 21201, USA. [3] Malaria Molecular Epidemiology Unit, Pasteur Institute in Cambodia, PO Box 983 Phnom Penh, Cambodia. [4] Laboratory of Malaria and Vector Research, NIAID, NIH, 12735 Twinbrook Parkway, Bethesda, MD 20852, USA. [5] Cochin Institute INSERM U1016, University Paris Descartes, and Laboratory of Parasitology-Mycology, Cochin Hospital, 22 rue Méchain, 75014 Paris, France. [6] Malaria Genetic and Resistance Group, Biology of Host-Parasite Interactions Unit, Pasteur Institute, 25-28 Rue du Dr. Roux, 75724 Paris, France. [7] Division of Infectious Diseases, Department of Medicine, Columbia University Irving Medical Center, New York, NY 10032, USA. [8] Present address: Parasites and Microbes, Wellcome Sanger Institute, Wellcome Genome Campus, Hinxton, Cambridge CB10 1SA, UK. [9] Present address: Parasitology and International Programs Branch, DHHS, NIAID, NIH, Rockville, MD 20852, USA. Correspondence and requests for materials should be addressed to D.A.F. (email: df2260@cumc.columbia.edu)

Over half of the world's population is at risk for infection with malaria parasites[1]. The World Health Organization estimates that in 2016 there were ~216 million cases of this mosquito-transmitted disease, resulting in 445,000 deaths[2]. The African continent bore an estimated 90% of all malaria morbidity and mortality, with most deaths occurring in children under the age of 5. In Africa, *Plasmodium falciparum* asexual blood stage (ABS) parasites are the primary cause of disease. Symptoms can include fever, severe malarial anemia, lactic acidosis, respiratory distress, or coma. Malaria caused by *P. falciparum* or *P. vivax* also afflicts Southeast (SE) Asia, the Indian subcontinent, the western Pacific, and South and Central America. The widespread use of artemisinin-based combination therapies (ACTs), combined with increased *Anopheles* mosquito vector control, has decreased the global malaria burden by an estimated 37% between 2000 and 2015[3]. These gains, however, are threatened by the rise of *P. falciparum* resistance to ACTs[1,4].

ACTs combine a fast-acting derivative of the endoperoxide artemisinin (ART) with a longer-acting partner drug. Current partners include the bis-4-aminoquinoline piperaquine (PPQ, comprising two chloroquine (CQ)-like moieties tethered by a central linker), the 4-aminoquinoline amodiaquine (ADQ), and the arylaminoalcohols mefloquine (MFQ) and lumefantrine (LMF)[4] (Supplementary Fig. 1). Multiple studies show diminishing efficacy for some ACTs in SE Asia[5–9]. Of greatest concern is the rapid spread of resistance to the ACT dihydroartemisinin (DHA) + PPQ[10–12]. This combination earlier showed superior therapeutic efficacy and post-treatment prophylaxis against drug-resistant *P. falciparum* and *P. vivax* malaria, and has been adopted as first-line therapy throughout much of SE Asia[13].

SE Asia has long been fertile ground for the development of antimalarial resistance. This region earlier saw the former first-line therapies CQ and sulfadoxine + pyrimethamine succumb to resistance, which then migrated into Africa[14,15]. The failure of CQ had devastating consequences in Africa, with substantial increases in mortality rates[3]. CQ resistance results from multiple mutations in PfCRT (*P. falciparum* chloroquine resistance transporter), which spans the membrane of the digestive vacuole (DV)[16]. These mutations enable PfCRT to efflux CQ out of the DV, thereby preventing CQ from binding to heme and inhibiting its detoxification. Heme is released upon enzymatic proteolysis of host hemoglobin (Hb), yielding globin chains that provide a vital source of amino acids for parasite growth. Studies have found Hb degradation and heme detoxification to be important for the modes of action of multiple antimalarials, including PPQ[4].

The emergence of *P. falciparum* resistance to ACT drugs in SE Asia was first reported in the late 2000s, when patients were observed in Cambodia to have extended parasite clearance times following treatment with the ART derivative artesunate[17]. ART resistance has been associated with mutant forms of the Kelch protein K13, first identified via in vitro DHA resistance selection and whole-genome sequencing of patient samples[18]. In vitro, ART resistance can be documented as elevated survival rates following exposure of newly invaded ring-stage parasites to a 6-h pulse of 700 nM DHA (the $RSA_{0-3 h}$ assay)[19]. $IC_{50}$ values do not substantially change in ART-resistant parasites[19]. A causal role for K13 mutations in conferring ART resistance has been confirmed using *k13* gene editing[20,21], and is supported by clinical and epidemiological studies[22–24]. Mutant *k13* alleles, particularly the relatively fit C580Y isoform, are now widespread throughout SE Asia[24,25]. Emerging resistance to the ART derivative results in a greater biomass of surviving parasites, which increases the opportunities for partner drug resistance to emerge.

In 2010, Cambodia officially adopted DHA + PPQ as the first-line therapy for uncomplicated *P. falciparum* malaria. This began in the western province of Pailin, and later was adopted throughout the country. The emergence of DHA + PPQ resistance was reported only a few years later in Cambodia[6,10,26] and in neighboring Vietnam[12]. As an example, in a clinical trial spanning 2012–14 in Oddar Meanchey Province, 54% of patients treated with DHA + PPQ recrudesced within 42 days[5]. In 2014, the official treatment recommendation in Cambodia was changed to artesunate + MFQ; however, in practice this only took effect 2–3 years later.

Resistance to PPQ has been difficult to assay in vitro, as resistant parasites often display unusual dose–response curves that are not sigmoidal and that often show incomplete killing regardless of drug concentration. These profiles complicate the use of standard $IC_{50}$ and $IC_{90}$ metrics to demonstrate resistance[11,27–29]. This observation led to the recent development of the Piperaquine Survival Assay ($PSA_{0-3 h}$) test, which uses survival rates to assess resistance[11]. Rates of ≥10% survival were associated with a 32-fold higher risk of recrudescence, establishing a working threshold of high-level in vitro resistance. Using the $PSA_{0-3 h}$, a recent study in Cambodia reported PPQ resistance in 23 of 31 isolates examined, consistent with high rates of local DHA + PPQ treatment failure[9].

Intense efforts to define the genetic basis of PPQ resistance suggest a multigenic trait that arises primarily on a K13 C580Y background[30]. Genome-wide association studies with SE Asian *P. falciparum* isolates point to several potential molecular markers, including amplification of *plasmepsins 2* and *3* (*pfpm2, pfpm3*) that contribute to Hb digestion in the DV[7,9,30], the Exonuclease1 E415G mutation[7,30], and several specific mutations in PfCRT[11,28].

*pfmdr1*, which like *pfcrt* encodes a DV membrane-resident putative transporter, has also been considered. Studies have revealed a reduced prevalence of multicopy *pfmdr1* since the implementation of DHA + PPQ[7,9,30,31], and an earlier report documented a mild inverse association between *pfmdr1* copy number and PPQ $IC_{50}$ values[32]. Other studies, including with isogenic parasites differing only in *pfmdr1* copy number, found no direct association with PPQ $IC_{50}$ values[27,33]. In the field, *pfmdr1* deamplification might also result from less use of MFQ, an ACT partner drug that selects for *pfmdr1* amplification. Reduced fitness of multicopy *pfmdr1* favors deamplification without MFQ pressure[34,35].

Here we test the hypothesis that novel alleles of *pfcrt*, recently detected in Cambodia[11,28], might represent an emerging mechanism of PPQ resistance. The premise included our recent demonstration that the PfCRT C101F mutation, selected through PPQ pressure and genetically edited into Dd2 parasites, mediated high-grade PPQ resistance in vitro[27]. We also earlier reported the emergence in French Guiana of a novel PfCRT C350R mutation that reduced parasite susceptibility to PPQ[36] in an area where DHA + PPQ is used. Recent data from Cambodia also identified a series of novel PfCRT mutations (H97Y, M343L, and G353V) in several isolates that were PPQ-resistant in the $PSA_{0-3 h}$[11]. A separate study also associated the F145I mutation with a five-fold increased risk of DHA + PPQ treatment failure[28]. Our results, obtained with *pfcrt*-modified parasites engineered in the Dd2 line (originating decades ago from a SE Asian isolate prior to ACT use), as well as with recent Cambodian isolates, provide compelling evidence that novel mutations in PfCRT represent a robust path to PPQ resistance. These findings illustrate the need to survey the emergence of novel PfCRT mutations in clinical isolates to ascertain their association with PPQ treatment failures in patient populations.

## Results

**Emergence of novel PfCRT mutations in Cambodia.** To examine PfCRT haplotypes in SE Asia, we downloaded

**Table 1 Haplotypes of *pfcrt*-edited and control *P. falciparum* lines**

| Parasite line | Edited *pfcrt* | 72 | 74 | 75 | 76 | 97 | 144 | 145 | 148 | 194 | 220 | 271 | 326 | 333 | 343 | 353 | 356 | 371 |
|---|---|---|---|---|---|---|---|---|---|---|---|---|---|---|---|---|---|---|
| | | \multicolumn PfCRT haplotype at listed position | | | | | | | | | | | | | | | | |
| Dd2 | No | C | *I* | *E* | *T* | H | A | F | L | I | *S* | *E* | *S* | T | M | G | *T* | *I* |
| Dd2^Dd2 crt | Yes | C | *I* | *E* | *T* | H | A | F | L | I | *S* | *E* | *S* | T | M | G | *T* | *I* |
| Dd2^Dd2 crt F145I | Yes | C | *I* | *E* | *T* | H | A | I | L | I | *S* | *E* | *S* | T | M | G | *T* | *I* |
| Dd2^Dd2 crt M343L | Yes | C | *I* | *E* | *T* | H | A | F | L | I | *S* | *E* | *S* | T | L | G | *T* | *I* |
| Dd2^Dd2 crt G353V | Yes | C | *I* | *E* | *T* | H | A | F | L | I | *S* | *E* | *S* | T | M | V | *T* | *I* |
| PH1008-C | No | C | *I* | *E* | *T* | H | A | F | L | I | *S* | *E* | *S* | T | L | G | *T* | *I* |
| PH1008-C^Dd2 crt | Yes | C | *I* | *E* | *T* | H | A | F | L | I | *S* | *E* | *S* | T | M | G | *T* | *I* |
| PH1263-C | No | C | *I* | *E* | *T* | Y | A | F | L | I | *S* | *E* | *S* | T | M | G | *T* | *I* |
| PH1263-C^Dd2 crt | Yes | C | *I* | *E* | *T* | H | A | F | L | I | *S* | *E* | *S* | T | M | G | *T* | *I* |
| Cam734 | No | C | *I* | **D** | *T* | H | **F** | F | **I** | **T** | *S* | *E* | N | **S** | M | G | I | R |
| GB4 | No | C | *I* | *E* | *T* | H | A | F | L | I | *S* | *E* | N | T | M | G | I | *I* |
| 3D7 | No | C | M | N | K | H | A | F | L | I | A | Q | N | T | M | G | I | R |

*pfcrt* was edited using customized zinc finger-nucleases, and clones were generated by limiting dilution. Differences from the 3D7 wild-type allele are shown in italics for Dd2 and bold for Cam734 mutations. Underline indicates mutations identified from piperaquine-resistant field isolates and tested herein by gene editing

whole-genome sequence data, generated by the *Plasmodium falciparum* Community Genome Project, from Asian *P. falciparum* isolates. Most samples were collected in 2011–12, with additional Cambodian samples from 2009–10 and 2013 (Supplementary Fig. 2)[37]. *pfcrt* coding sequences were then assembled to identify non-synonymous point mutations. Our analysis resulted in 869 genomes with high-confidence *pfcrt* full-length coding sequences. 59% of these samples came from Cambodia, while the others came from Laos, Myanmar, Thailand, Vietnam, and Bangladesh (Supplementary Table 1).

We identified the Dd2, Cam734, and GB4 PfCRT haplotypes (Table 1) as the most common through 2009–13, with the 8-amino acid variant Dd2 haplotype constituting 58% of all sequenced samples (Fig. 1a; Supplementary Tables 1, 2). The 9-amino acid variant Cam734[38] was present at 14%. The 6-amino acid variant GB4 haplotype, which is quite prevalent in Asia and Africa[39], was present at 13%. Only 0.8% of isolates harbored the canonical wild-type (3D7) haplotype. The vast majority of novel PfCRT mutations were found to have evolved individually on the Dd2 haplotype, almost exclusively in Cambodia (Fig. 1a; Supplementary Table 1). These mutations included H97Y, F145I, M343L, and G353V (Table 1).

To further investigate *pfcrt* allelic diversity in Cambodia, we analyzed the *P. falciparum* genome sequences from an additional 93 patient isolates collected by researchers at the Pasteur Institute in Cambodia from several sites throughout western Cambodia during 2010–16 (Fig. 1b). These results provided evidence of a substantial increase in the prevalence of novel PfCRT variants over time, with zero variants observed in 2010 (0 of 8) increasing to 95% (20 of 21) in the 2016 cohort. In 2012–13 the predominant mutations were M343L and G353V. By 2016 the most common variant was G353V (12 of 21). In Pursat, all 19 isolates collected in 2016 harbored novel PfCRT mutations (Supplementary Table 3). Intriguingly, of the 34 isolates harboring a novel PfCRT mutation, all were on the Dd2 PfCRT background and all carried K13 C580Y. In contrast, parasites expressing the PfCRT Dd2 haplotype without additional mutations were a mixture of mutant (C580Y and R539T) and wild-type K13. Parasites expressing the less prevalent GB4 and Cam783 PfCRT haplotypes were all K13 wild-type (Supplementary Table 3).

**PPQ resistance in Cambodian isolates with variant PfCRT.** We tested 51 of these Pasteur Cambodian isolates from 2010 to 2013 in the PSA$_{0-3 h}$. Results showed a strong association between the

PfCRT H97Y, M343L, or G353V variants and increased PPQ survival. Each PfCRT variant isolate (colored in magenta in Fig. 1c–e) was found to be PPQ-resistant (survival rates > 10%; Fig. 1c). Other isolates from that period were PPQ-resistant by the PSA$_{0-3 h}$ but did not contain any of these PfCRT mutations. Multicopy *pfpm2* copy number was positively associated with increased PPQ survival, although several *pfpm2* multicopy parasites were PSA-sensitive (Fig. 1d). Each of the PfCRT variants carrying H97Y, M343L, or G353V was observed on a background with three or more *pfpm2* copies. These variants were also exclusively single-copy *pfmdr1* (Fig. 1e). All PPQ-resistant parasites were single copy for *pfmdr1*, in comparison to PPQ-sensitive parasites that carried 1–4 *pfmdr1* copies.

**Gene-edited PfCRT variants mediate PPQ resistance.** To test the role of these novel PfCRT mutations in PPQ susceptibility, we genetically edited[40] these into a standard laboratory parasite line (Dd2) or removed them from PPQ-resistant Cambodian isolates (Supplementary Fig. 3a). We introduced the F145I, M343L, or G353V mutations into Dd2, as confirmed by PCR and sequencing of the recombinant locus (Supplementary Fig. 3b). Edited parasites were cloned by limiting dilution to generate Dd2^Dd2 crt F145I, Dd2^Dd2 crt M343L, and Dd2^Dd2 crt G353V (Table 1). As a recombinant control, we also used the recombinant isogenic line Dd2^Dd2 crt, which expresses the Dd2 *pfcrt* allele[40]. The location of these mutations in the predicted 10-transmembrane domain organization of PfCRT is illustrated in Supplementary Fig. 3c.

We also obtained the Cambodian patient isolates PH1008-C and PH1263-C (collected in 2012 and 2013, respectively) that harbor the novel M343L and H97Y mutations, respectively, on the PfCRT Dd2 haplotype background. Both isolates are single-copy *pfmdr1* and carry K13 C580Y. These mutations were removed by gene editing, yielding the recombinant clones PH1008-C^Dd2 crt and PH1263-C^Dd2 crt, respectively (Table 1). Culture-adapted Cambodian isolates harboring PfCRT G353V or F145I mutations were unavailable for these experiments.

We next assayed our panel of parasites to determine the effect of PfCRT mutations on PPQ survival in the PSA$_{0-3h}$ (Fig. 2), as well as on IC$_{50}$ values for PPQ and other antimalarial drugs (Figs. 3, 4). Results were grouped into either the addition of novel single nucleotide polymorphisms (SNPs) into Dd2 or the removal of novel SNPs from the Cambodian isolates.

The addition of PfCRT SNPs to Dd2 parasites had a dramatic effect on PPQ survival, as measured using the PSA$_{0-3 h}$. Assays with the PPQ-sensitive lines 3D7, GC03, Dd2, and the edited self-

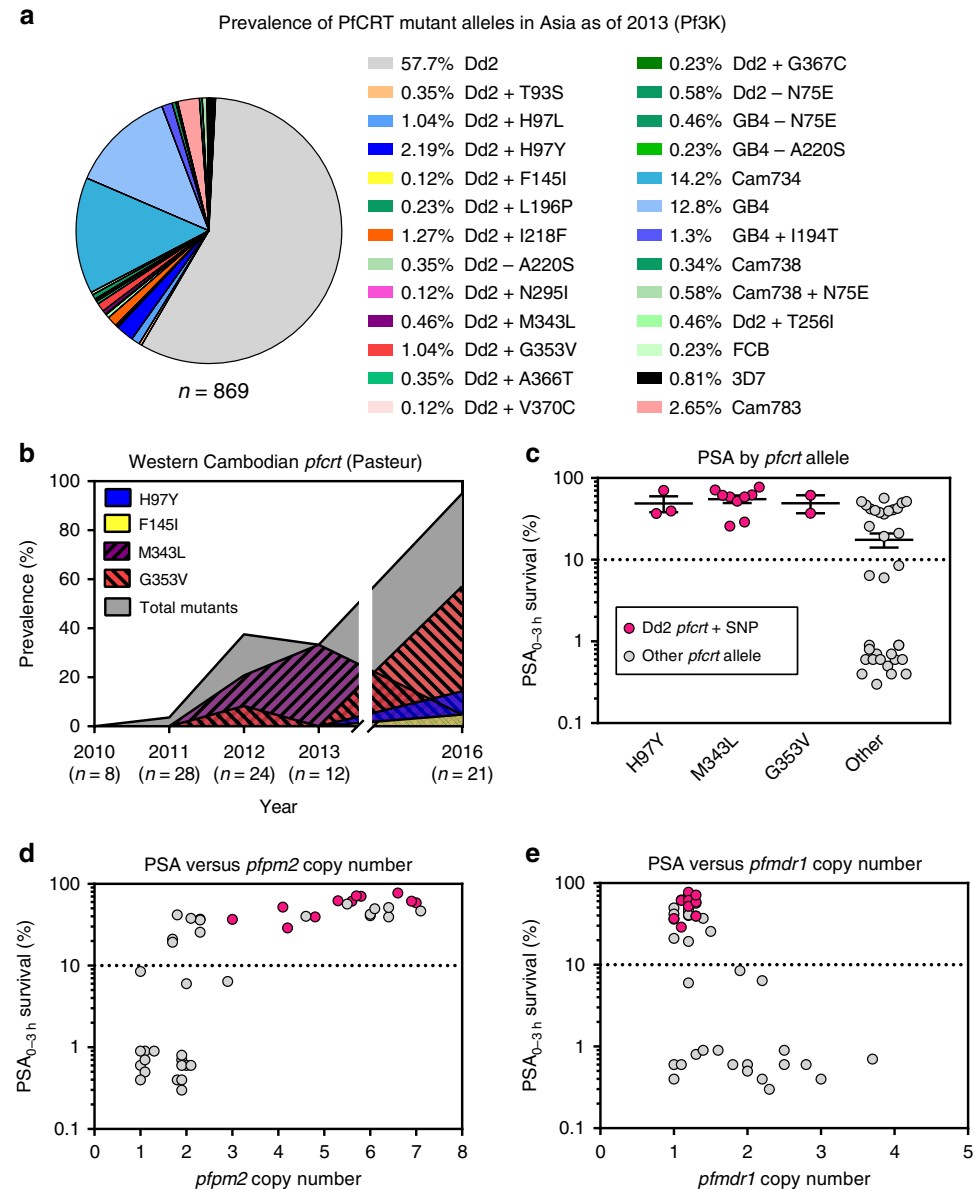

**Fig. 1** Emergence of novel PfCRT mutations in SE Asia that associate with elevated $PSA_{0-3h}$ survival. **a** Sequence analysis of 869 *P. falciparum* genomes from multiple Asian countries, generated by the MalariaGEN consortium Pf3K database[37], identified three major PfCRT haplotypes (Dd2, Cam734, GB4) and the emergence of multiple variants at low prevalence. Haplotypes and their distribution by country are listed in Supplementary Tables 1, 2. Most isolates were collected in 2011–12, with additional Cambodian samples from 2009 to 2010 and 2013 (Supplementary Fig. 2). **b** Genome analysis of a separate set of 93 western Cambodian isolates, collected over the period 2010–16 by the Pasteur Institute in Phnom Penh, Cambodia (Supplementary Table 3), provided evidence of a rapid increase in the prevalence of novel PfCRT variants on the Dd2 allelic background. A subset of 51 of these Pasteur Cambodian isolates from the period 2010–13 were also typed for $PSA_{0-3h}$ survival rates and *pfpm2* and *pfmdr1* copy numbers, as shown in **c–e**. **c** Isolates harboring mutations H97Y, M343L, or G353V were associated with elevated $PSA_{0-3h}$ survival rates, an indicator of PPQ resistance (F145I had not been detected at that time). PPQ-resistant isolates were also observed that lacked these specific mutations. The dashed line of 10% survival represents a previously designated resistance threshold associated with a 32-fold increased risk of clinical failure[11]. **d** All PPQ-resistant parasites in this sample set had multiple copies of *pfpm2*, although some parasites with multiple copies were also in the PPQ-sensitive survival range. All parasites harboring one of the three PfCRT mutations in **c** and **d** (designated in magenta) were multicopy *pfpm2*. **e** All PPQ-resistant parasites observed were single-copy *pfmdr1*. Magenta samples harbored novel PfCRT mutations represented in **b**

replaced $Dd2^{Dd2\ crt}$ line showed <1% survival relative to their vehicle-treated control (Fig. 2; Supplementary Table 4). In contrast, each added PfCRT mutation conferred a mean survival > 10%, indicative of in vitro resistance[11]. Edited lines expressing the F145I, M343L, or G353V mutations introduced into the Dd2 haplotype yielded mean survival rates of 12.4–23.4%, with the F145I mutation conferring the highest level of survival (Fig. 2). The inverse experiment, removing novel

PfCRT mutations from field isolates, resulted in the complete loss of PPQ resistance. The PH1008-C isolate, with a Dd2 + M343L *pfcrt* allele, showed an average of 14.4% PPQ survival. This elevated survival was fully ablated by removing the M343L SNP. Similarly, the PH1263-C isolate, with a Dd2 + H97Y *pfcrt* allele, had an average of 14.5% survival. This survival was also ablated upon removal of the H97Y SNP (Fig. 2). These results provide

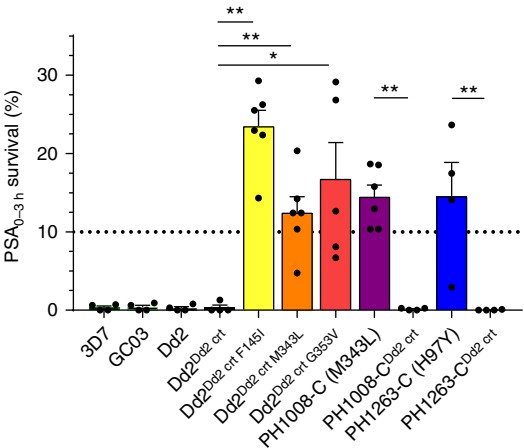

**Fig. 2** The PfCRT mutations H97Y, F145I, M343L, and G353V confer PPQ resistance. When edited into to a PPQ-sensitive Dd2 *pfcrt* background, F145I, M343L, and G353V each conferred PPQ resistance, as determined using the in vitro PSA$_{0-3h}$. In this PPQ survival assay, tightly synchronized ring-stage parasites were exposed to 200 nM PPQ for 48 h, followed by removal of drug and further culture incubation for 24 h. Survival rates were determined as a percentage of the parasitemia obtained with each line mock-treated with drug vehicle, and are presented as means ± SEM. When H97Y or M343L were removed from the contemporary Cambodian field isolates PH1008-C and PH1263-C, PPQ resistance was fully ablated. N, n = 2–4, 2 (Supplementary Table 4). Significance was determined using Mann–Whitney *U* tests. Comparisons are shown between *pfcrt*-edited parasites and their isogenic controls. *$p < 0.05$; **$p < 0.01$

evidence that these four mutations can each mediate in vitro resistance to PPQ.

Conventional 72-h dose–response assays revealed complex profiles with PPQ-resistant parasites, as previously reported[9,11,29]. For Dd2 parasites, the addition of F145I conferred significantly higher mean PPQ IC$_{50}$ and IC$_{90}$ levels, whereas the G353V mutation significantly increased only the IC$_{90}$ level (Fig. 3a, b; Supplementary Table 5). With both mutations, IC$_{90}$ values exceeded 3 μM, a vast increase over the 50 nM IC$_{90}$ observed with the Dd2$^{Dd2\ crt}$ isogenic control line. Dose–response profiles illustrated these exceptionally high IC$_{90}$ values and showed incomplete growth inhibition across a wide range of PPQ concentrations in both the Dd2$^{Dd2\ crt\ F145I}$ and Dd2$^{Dd2\ crt\ G353V}$ lines. Another phenotype was observed with the M343L mutation, which showed considerably smaller, albeit significant IC$_{90}$ increases (Figs. 3b, c). The highly unusual dose–response curves observed with the two parental Cambodian isolates PH1008-C and PH1263-C reverted to sigmoidal curves typical of sensitive parasites following the removal of the M343L and H97Y mutations, respectively (Fig. 3d, e).

**PPQ uptake is dependent on the parasite genetic background.** To examine the role of PfCRT mutations on PPQ accumulation in our parental versus edited cell lines, we measured [³H]-PPQ uptake into synchronized trophozoites. Cultures were incubated with 10 nM [³H]-PPQ at 37 °C for 1 h and radioactivity was measured separately for cells and supernatants. From this, we derived the cellular accumulation ratio (CAR), i.e., the ratio of intracellular versus extracellular [³H]-PPQ (Fig. 3f; Supplementary Table 6). The most significant difference in the CAR was observed between cell lines expressing the CQ- and PPQ-sensitive 3D7 *pfcrt* allele and all others expressing the CQ-resistant Dd2 *pfcrt* allele or its PPQ-resistant variants. Compared to 3D7, the CAR was ~two-fold and ~two to four-fold less in the Dd2 and Cambodian

variants, respectively. These data suggest that the wild-type 3D7 PfCRT isoform allows for increased PPQ accumulation in the DV, where retention is predicted to be driven by the weak-base gradient and PPQ interactions with heme and hemozoin[27]. Notably, there was no significant difference between the edited Dd2$^{Dd2\ crt}$ PPQ-sensitive control line versus the Dd2 PPQ-resistant variants expressing the PfCRT mutations F145I, M343L, or G353V, suggesting that these mutations do not confer resistance by accumulating less PPQ in the Dd2 background. Interestingly, there was a ~1.8-fold increase in the CAR for the edited recombinant clones PH1008-C$^{Dd2\ crt}$ and PH1263-C$^{Dd2\ crt}$, which were sensitized to PPQ, compared with their parental PPQ-resistant isolates. The influence of the genetic background on PPQ uptake in PPQ-resistant versus sensitive variants highlights the complex relationship between these PfCRT mutations and PPQ resistance, which does not appear to be explained solely by changes in drug accumulation or efflux.

**Novel PfCRT mutations impact multiple antimalarial drugs.** For CQ and its primary metabolite md-CQ, we observed strikingly different impacts (Fig. 4a, b; Supplementary Table 7). In the Dd2 background, the addition of each of the three PfCRT mutations F145I, M343L, or G353V attenuated the CQ resistance phenotype of the parental Dd2 line, despite the presence of the K76T mutation. The greatest impact was observed with the F145I mutant, with 6.7- and 8.6-fold reductions in mean IC$_{50}$ values for CQ and md-CQ, respectively. With PH1008-C, removing M343L lowered the IC$_{50}$ value for CQ but not for md-CQ. Attenuation of CQ and md-CQ resistance was also observed upon removal of the H97Y mutation in the PH1263-C line, indicating that in these parasites H97Y contributed to resistance to CQ and its metabolite. Thus, the gain of PPQ resistance afforded by the introduction of these novel PfCRT mutations was generally accompanied by increased sensitization to CQ and md-CQ in a genetic background-dependent manner.

A similar pattern of altered susceptibilities was also observed with the structurally related drug ADQ, tested here with its active metabolite md-ADQ (Fig. 4c; Supplementary Table 8). Again, the introduction of the F145I mutation afforded the greatest reduction in IC$_{50}$ values in Dd2, whereas the H97Y mutation contributed to md-ADQ resistance in the PH1263-C line. Lower md-ADQ IC$_{50}$ values were also observed upon adding M343L or G353V to Dd2 parasites; however, removing M343L did not significantly affect md-ADQ in the PH1008-C line. These data highlight a role for other loci in determining the impact of PfCRT mutations on both CQ and md-ADQ susceptibility.

In the Dd2 background, QN followed a similar pattern as CQ and ADQ, with the addition of F145I, M343L, or G353V each resulting in significantly lower IC$_{50}$ values (Fig. 4d; Supplementary Table 8). Paradoxically, the M343L mutation significantly increased the QN IC$_{50}$ value in PH1008-C parasites. In the PH1263-C line, removal of H97Y resulted in a gain of QN resistance, i.e., opposite to the effect of this mutation on CQ and md-ADQ susceptibility. This complex interplay between *pfcrt* and the genetic background, whereby PfCRT mutations can dictate whether changes in QN responses track with those of CQ and ADQ, or go in opposite directions, has been previously observed[41–43]. In the Cambodian isolates, the impacts of M343L and H97Y on QN IC$_{50}$ values were similar for MFQ and LMF (Fig. 4e, f; Supplementary Table 9). Small but significant changes in IC$_{50}$ values for DHA were also observed for some mutations (Supplementary Table 10).

**CQ and PPQ verapamil reversibility differs.** We also explored the effects of verapamil (VP), a CQ resistance reversal agent[44], on

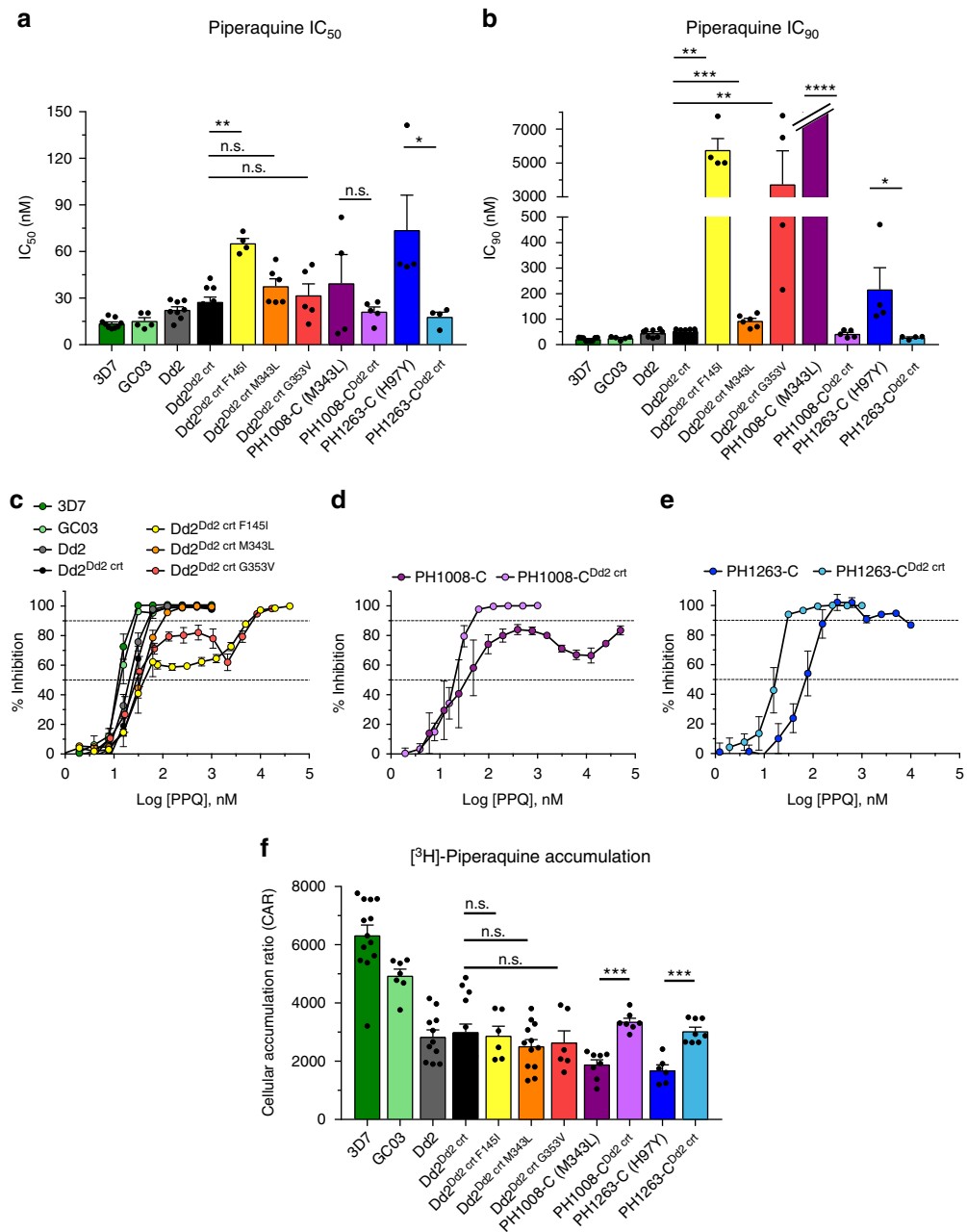

**Fig. 3** PPQ dose–response and accumulation data illustrate distinct variant PfCRT-mediated PPQ resistance profiles. **a** Mean ± SEM $IC_{50}$ values were determined from conventional 72-h dose–response assays performed with asynchronous parasite cultures. **b** Mean ± SEM $IC_{90}$ data from the same assays. Note that PH1008-C never attained ≥90% inhibition at 50 μM (Supplementary Table 5) and is nominally listed as 7 μM. **c–e** Dose–response curves for Dd2, PH1008-C, and PH1263-C reference lines and *pfcrt*-edited clones. The PPQ-sensitive 3D7 and GC03 lines are shown alongside the collection of Dd2 lines as additional sensitive controls. Data show mean ± SEM percent growth inhibition as a function of [PPQ]. These results show the wide variability in dose–response profiles observed with the most highly resistant lines. **f** Cellular accumulation ratios (CAR), representing the ratio of intracellular to extracellular [³H]-PPQ after 1 h incubation at 37 °C. Assays employed synchronized trophozoites, when the DV is fully formed and hemoglobin degradation is maximal. CAR results for each line are presented as means ± SEMs (N, n = 3 to 7, 2; Supplementary Table 6). **a**, **b**, **f** Significance was determined using Mann–Whitney U tests comparing *pfcrt*-edited parasites and their isogenic controls. *$p < 0.05$; **$p < 0.01$; ***$p < 0.001$; ****$p < 0.0001$; n.s. not significant

CQ and PPQ susceptibility. For CQ, we observed a ~72% reduction in $IC_{50}$ values in the presence of 0.8 μM VP for Dd2 and Dd2$^{Dd2\ crt}$, consistent with earlier reports[41,45]. The addition of the F145I, M343L, or G353V mutations led to significantly less reversibility (26–53%) (Supplementary Table 11). No reversibility was observed with the CQ-sensitive 3D7 and GC03 lines. In the parental and edited Cambodian lines, VP reversibility was observed at 60–65%, irrespective of their PfCRT haplotype or *pfpm2* copy number. With PPQ, we observed 17–43%

reversibility among the PPQ resistance-conferring variants on the Dd2 background; however, these differences were not significant compared to the Dd2$^{Dd2\ crt}$ isogenic control. PPQ-sensitive lines displayed minimal VP reversibility (2–17%). Two factors complicated our analysis of the effect of VP on PPQ $IC_{50}$ values. First, 0.8 μM VP on its own inhibited growth by 18–45% in the lines expressing a novel PfCRT mutation (H97Y, F145I, M343L, or G353V), compared to <14% inhibition for all other lines (Supplementary Table 11). Second, dose–response profiles were quite

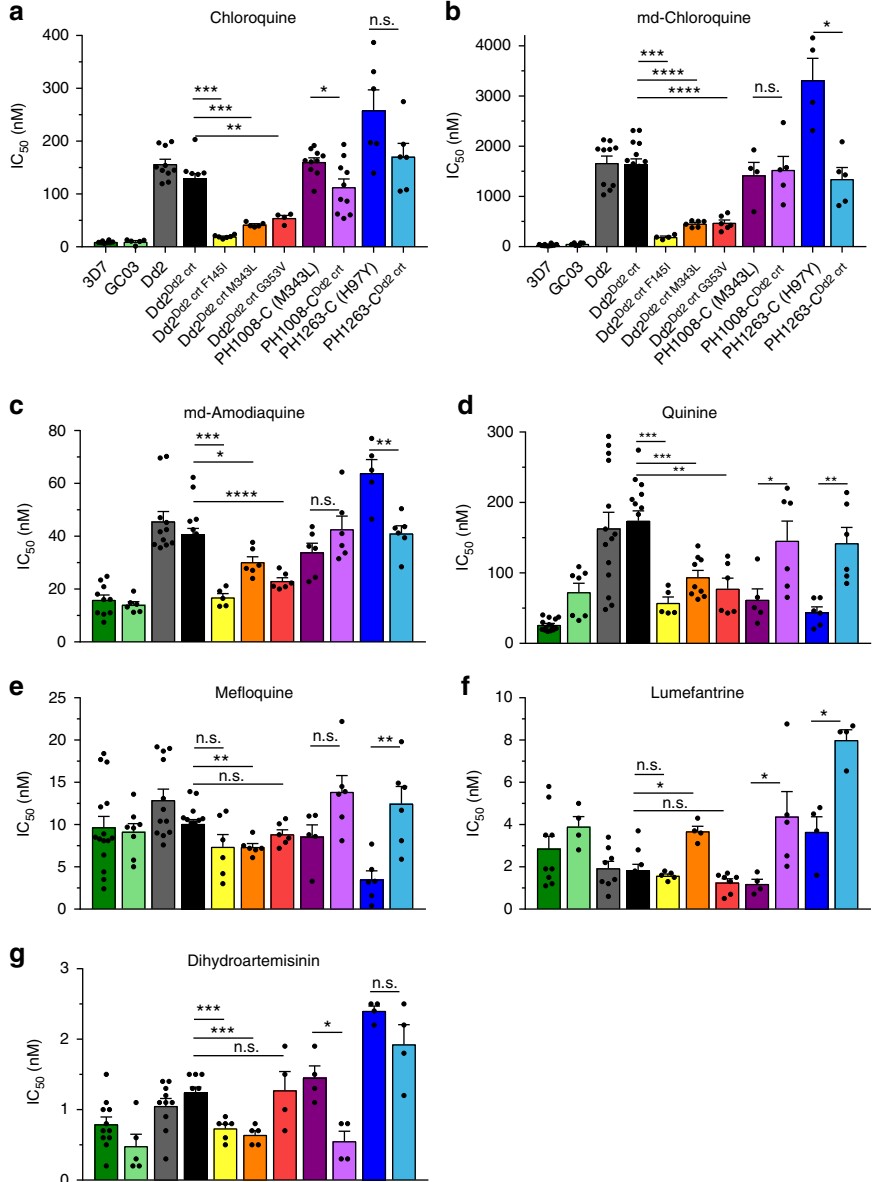

**Fig. 4** Emerging PfCRT mutations modify *P. falciparum* susceptibility to multiple antimalarial drugs. Mean ± SEM $IC_{50}$ values (Supplementary Tables 7–10) were calculated from 72-h dose–response assays for drugs designated in **a**–**g**. *N*, *n* = 4+, 2. Significance was determined using Mann–Whitney *U* tests comparing *pfcrt*-edited parasites and their isogenic controls. *p < 0.05; **p < 0.01; ***p < 0.001; ****p < 0.0001; n.s. not significant, md monodesethyl

flat around the region of 50% inhibition, creating sizable variation (Fig. 3).

$PSA_{0-3 h}$ tests also revealed no significant impact of 0.8 μM VP on percent survival of either $Dd2^{Dd2 \ crt \ F145I}$, which had the highest mean survival, or the PPQ-sensitive lines $Dd2^{Dd2 \ crt}$, Dd2, 3D7, or GC03 (Supplementary Table 4). Taken together, these results provide evidence that VP does not substantially reverse PPQ resistance, unlike with CQ, highlighting mechanistic distinctions between PfCRT-mediated CQ and PPQ resistance.

**Novel PfCRT mutations differentially impact ABS growth**. Our data with *pfcrt*-modified lines provided evidence that emerging Cambodian PfCRT mutations can confer in vitro resistance to PPQ. In vivo infections are more complex, requiring a balance between resistance and fitness. In *P. falciparum*, fitness broadly includes the capacity for sexual development and transmission as well as the ABS growth rate under different conditions of host

nutrition, immunity, and antimalarial drug presence. As a proxy for ABS fitness, we co-cultured our test lines with an eGFP-expressing reference line and measured the proportion of eGFP+ parasites over 20 days (10 generations) (Fig. 5; Supplementary Table 12).

The eGFP+ line was a modified Dd2, so we expected Dd2 to have slightly better fitness due to the lack of the eGFP burden. Indeed, $Dd2^{Dd2 \ crt}$ mildly outcompeted eGFP+ parasites (Fig. 5a; Supplementary Table 12). The $Dd2^{Dd2 \ crt}$ and $Dd2^{Dd2 \ crt \ M343L}$ curves were very similar, indicating a relatively small effect of M343L on ABS fitness. $Dd2^{Dd2 \ crt \ F145I}$ and $Dd2^{Dd2 \ crt \ G353V}$, on the other hand, were much less fit, and were quickly outcompeted by eGFP+ parasites.

Both unmodified parental field isolates were strongly outcompeted by the eGFP+ line in as few as two generations, indicating that these isolates were much less fit. Removing the novel PfCRT mutations rescued the growth defect, and both

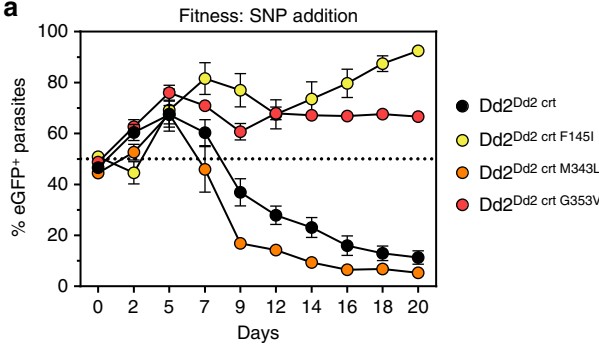

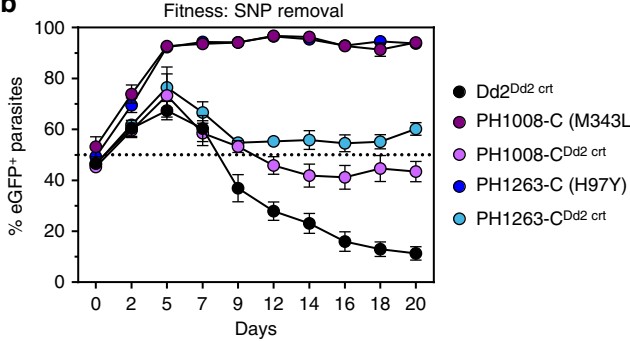

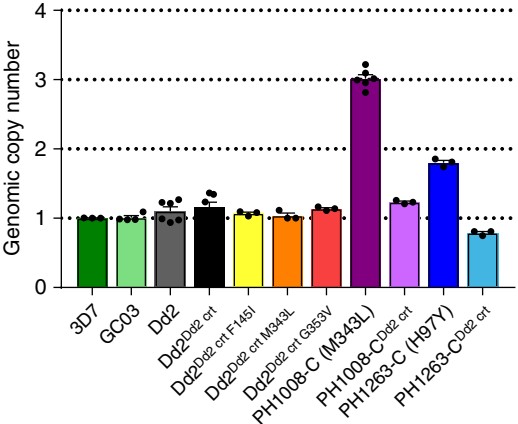

**Fig. 6** *pfpm2* amplification is not essential for PPQ resistance but may be associated. qPCR-based determination of *pfpm2* copy number, shown as means ± SEM. Results showed the presence of only one *pfpm2* copy in PPQ-resistant genetically edited Dd2 lines that expressed novel *pfcrt* alleles. The multiple *pfpm2* copies observed in the Cambodian lines PH1008-C and PH1263-C were reduced to single copies upon removal of the PfCRT M343L or H97Y mutations. These data were consistent with the decrease in RNA expression levels observed by RT-qPCR (Supplementary Table 13)

**Fig. 5** PfCRT mutations mediate different impacts on *P. falciparum* growth rates in vitro. *pfcrt*-edited lines were co-cultured for 20 days with a GFP-expressing Dd2 line and cultures were sampled every 2–3 days by flow cytometry. Experiments were conducted on two separate occasions in triplicate, and the mean ± SEM percentage of GFP+ parasites were plotted over time. Values below the 50% dashed line indicate that a line was fitter than GFP+ parasites. **a** Results with PfCRT SNPs added to the Dd2 PfCRT haplotype in recombinant Dd2 parasites. Only the M343L mutation had comparable fitness to the Dd2 allele, whereas the two other variants (F145I and G353V) were less fit. **b** Results of Cambodian isolates and isogenic lines with novel PfCRT SNPs removed, showing improved fitness in the lines lacking H97Y or M343L. Raw data are presented in Supplementary Table 12

PH1263-C$^{Dd2\ crt}$ and PH1008-C$^{Dd2\ crt}$ remained at approximately a 1:1 ratio with the eGFP+ line throughout the 10 generations. Removing M343L from PH1008-C dramatically improved in vitro growth rates, whereas adding M343L to Dd2 had a small effect on growth. This observation, coupled with the disparate effect of adding or removing PfCRT mutations on CQ and md-CQ susceptibility, suggests an important contribution of the strain background.

**PfCRT can confer PPQ resistance without multicopy *pfpm2*.** In light of recent reports that multicopy *pfpm2* is a molecular marker of PPQ resistance[7,9,28], we investigated *pfpm2* copy number in our gene-edited and parental parasites using quantitative PCR (qPCR)[9]. This analysis indicated a single copy of *pfpm2* in all *pfcrt*-modified lines generated from Dd2 (Fig. 6; Supplementary Table 13). Transcript levels, measured by reverse transcription quantitative PCR (RT-qPCR), were also unchanged (Supplementary Table 13).

***pfpm2* is deamplified in *pfcrt*-edited Cambodian lines.** Our PCR and qRT-PCR studies provided evidence that PH1008-C (M343L) and PH1263-C (H97Y) had three and two copies of *pfpm2*, respectively, unlike the single-copy Dd2. This observation is consistent with the frequent presence of multicopy *pfpm2* in

PPQ-resistant Cambodian isolates (Fig. 1d)[7,9,28]. Intriguingly, our qPCR assays detected only a single copy of *pfpm2* in our *pfcrt*-edited PH1008-C$^{Dd2\ crt}$ and PH1263-C$^{Dd2\ crt}$ clones. Loss of the additional copies was confirmed by RT-qPCR data (Supplementary Table 13).

**Novel PfCRT mutations edited into Dd2 create distended DVs.** During the culturing of our parental and *pfcrt*-edited lines, we observed a distended and translucent DV phenotype during the development from mid-trophozoites to mid-schizonts, which was specific to the *pfcrt*-edited lines Dd2$^{Dd2\ crt\ F145I}$, Dd2$^{Dd2\ crt\ M343L}$, and Dd2$^{Dd2\ crt\ G353V}$ (Fig. 7). The recombinant control line Dd2$^{Dd2\ crt}$ and the parental Dd2 did not display this phenotype. This trait was also not observed in the PPQ-resistant Cambodian lines PH1008-C or PH1263-C or in their recombinant derivatives PH1008-C$^{Dd2\ crt}$ and PH1263-C$^{Dd2\ crt}$ that expressed the Dd2 *pfcrt* allele.

**Discussion**

The recent evolution in Cambodia of *P. falciparum* resistance to PPQ, building on the earlier emergence of ART resistance, has rapidly compromised the clinical efficacy of DHA + PPQ[9]. This situation, exacerbated by the current paucity of effective alternatives, poses a dire threat to the region, where DHA + PPQ has been widely adopted as first-line therapy[2]. Here, we report compelling evidence that emerging mutations in PfCRT can mediate high-level resistance to PPQ (Figs. 2 and 3). These results were generated using genetically edited lines in which a series of PfCRT mutations were either added to the PPQ-sensitive Dd2 line or removed from the PPQ-resistant Cambodian isolates PH1008-C and PH1263-C.

Our PSA$_{0-3\ h}$ assays showed >10% survival following PPQ exposure in Dd2 parasites expressing the PfCRT mutations F145I, M343L, or G353V, contrasting with essentially no survival in parasites expressing the parental Dd2 haplotype (Fig. 2). The >10% survival rates observed with the two Cambodian isolates were also reduced to effectively zero upon removal of H97Y or M343L. Strikingly, each mutation produced a unique dose–response curve, with three of the four

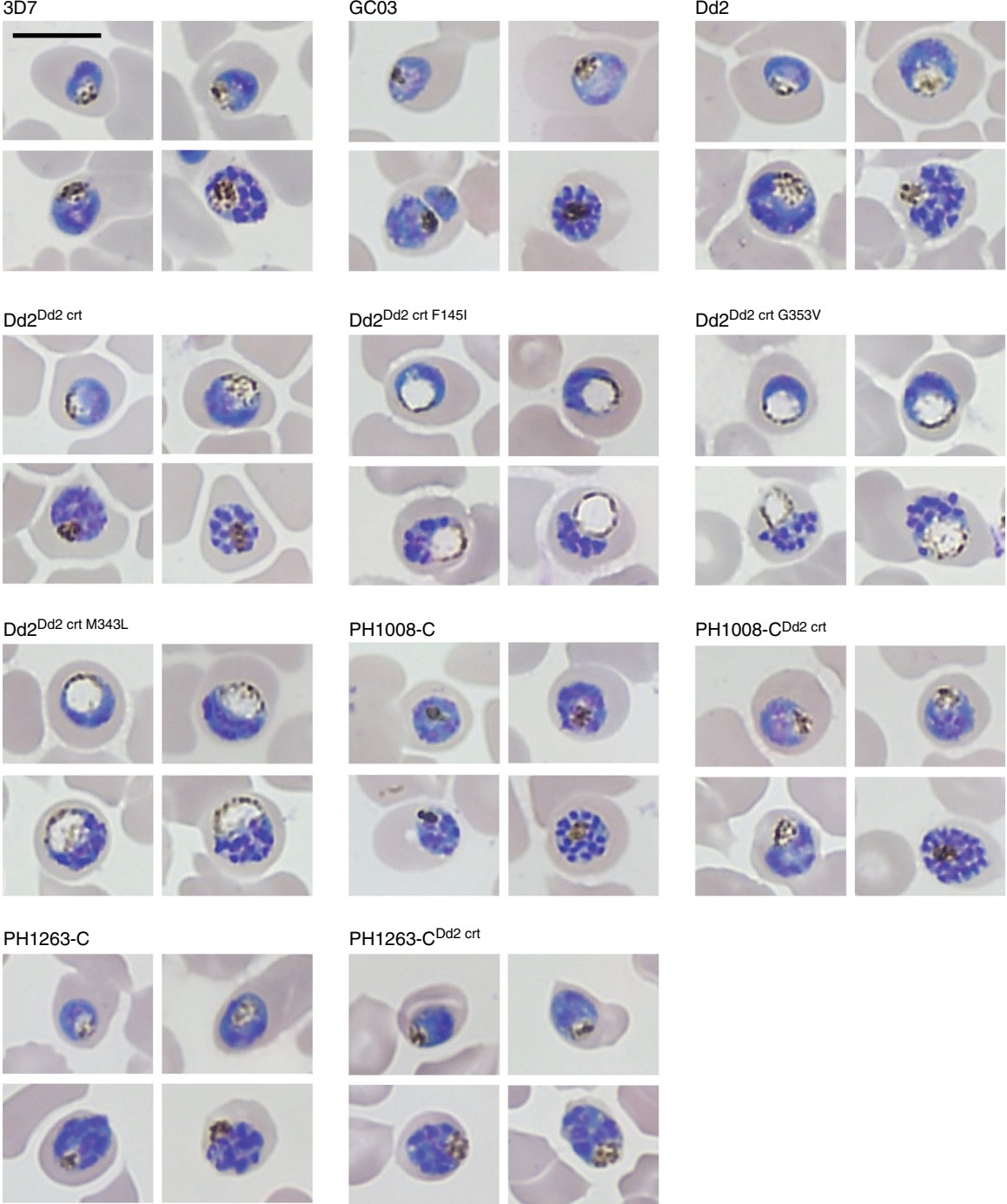

**Fig. 7** Cell morphology of *pfcrt*-edited and parental parasite lines. Images reveal distended, translucent digestive vacuoles in Dd2 parasites expressing novel PfCRT variants. These aberrant morphological features were not observed in recent Cambodian isolates harboring novel variants. The scale bar measures 10 μm

mutations (all but M343L) being associated with incomplete inhibition up to μM concentrations of drug (Fig. 3c–e). These data provide compelling evidence that novel PfCRT mutations can mediate PPQ resistance.

Our analysis of 869 Asian *P. falciparum* genomes sequenced by the Pf3K consortium provides evidence of a remarkable array of novel PfCRT mutations (Fig. 1; Supplementary Tables 1, 2). These emerged almost exclusively in Cambodia, which was the first country to adopt DHA + PPQ as first-line therapy. Analysis of an additional 93 Cambodian genomes collected by

investigators at the Pasteur Institute in Cambodia suggests a rapid increase in the prevalence of these variants (Supplementary Table 3). These results highlight the need to survey contemporaneous field isolates for the emergence of novel mutations and evaluate their potential role as biomarkers of DHA + PPQ treatment failure.

Our findings raise the question of why was PfCRT not identified as a molecular marker of PPQ resistance in the first two Cambodian genome-wide association studies[7,9]. One explanation could be that a proportion of PPQ-resistant isolates do not involve mutations in

PfCRT and instead rely on an alternative primary determinant. *pfpm2* is one such candidate. Indeed, half of our culture-adapted PPQ-resistant isolates appeared to not harbor novel PfCRT variants (Fig. 1c). Another factor could lie in the complex nature of the variant *pfcrt* sequences. Our analysis of Cambodian isolates revealed three major haplotypes (Dd2, Cam734, and GB4). Interestingly, the novel PfCRT mutations that we found to mediate PPQ resistance were observed only on the Dd2 haplotype (Supplementary Tables 1, 2), complicating association studies. Assembly and analysis of Illumina-based sequencing of the 13-exon *pfcrt* gene is also notoriously difficult. A third explanation might be that the emergence of PfCRT mutations is a recent and rapidly evolving phenomenon that was not captured by the two initial association studies. Witkowski et al.[9] examined whole-genome sequence data only from samples collected in 2012, whereas 80% of the Amato et al.[7] samples were from 2010 to 2012, with the remainder from 2013. Of note, the recent genome-wide study by Agrawal by et al.[28], which associated PfCRT F145I with PPQ resistance, first observed this mutation in samples from 2013 to 2014. Our data from 2016 revealed novel PfCRT variants in 20 of 21 isolates, dominated by G353V.

Deep sampling of contemporary SE Asian isolates will be particularly informative in determining whether certain PfCRT mutations gain dominance in Cambodia and from there potentially spread into neighboring areas of PPQ use. This scenario recalls the earlier emergence of multiple *k13* alleles in SE Asia that coalesced into a dominant C580Y haplotype that has since spread with a hard sweep across the region[22,24,31]. PPQ resistance appears to have rapidly emerged, predominantly in K13 C580Y parasites harboring multicopy *pfpm2*, and has been detected in western Cambodia, Thailand, and Vietnam[25,31].

It is well established that PfCRT isoforms giving rise to CQ resistance, such as Dd2, are able to transport CQ away from their primary site of action via a DV efflux mechanism, reducing intracellular drug accumulation in CQ-resistant strains[46–50]. We examined whether the novel PfCRT isoforms might similarly mediate PPQ transport, given the structural similarity between PPQ and CQ and the evidence that both drugs accumulate to high concentrations in the DV and inhibit β-hematin formation[51]. Indeed, our recent Hb fractionation studies found both drugs to be similar in their inhibition of hemozoin formation and the resulting buildup of the reactive heme precursor[27]. We found no significant difference in [³H]-PPQ accumulation between our edited Dd2 lines expressing the F145I, M343L, or G353V mutations compared to Dd2 *pfcrt*. With the Cambodian isolates, we observed only a relatively small increase in accumulation in the revertant PPQ-sensitive lines compared to the resistant parents. These results suggest that this variant PfCRT-mediated PPQ resistance mechanism is not primarily a result of reduced intracellular drug accumulation. One alternative explanation might involve PPQ-mediated binding to and functional inhibition of certain PfCRT isoforms, recalling earlier reports of distinct drug binding sites in this transporter[49,52,53]. Further investigations in gene-edited parasites and existing *Xenopus* oocyte- or yeast-based heterologous expression systems[50,53] will be important in delineating the relationship between PfCRT mutations, *pfpm2* copy number, drug accumulation, and PPQ resistance.

Additional evidence that PfCRT-mediated CQ and PPQ resistance differ mechanistically was obtained by analyzing the effect of VP, which strongly reversed CQ but not PPQ resistance. In the Dd2 background, the novel PfCRT mutations significantly reduced the degree of VP reversibility on CQ and also resulted in greater toxicity of VP alone, when compared with parasites expressing the Dd2 isoform. Unexpectedly, our four Cambodian parental and edited lines all displayed similar levels of VP reversibility for CQ, despite differences in their PfCRT haplotype and *pfpm2* copy number. These findings suggest that other loci in the Cambodian isolates might modify the VP response, which until now has only been linked to PfCRT[54].

Our in vitro competition assays revealed a substantial growth defect in Dd2 parasites expressing the PfCRT mutations F145I and G353V, but not M343L. These lines all displayed a striking morphological trait of having swollen, translucent DVs in maturing trophozoites and schizonts (Fig. 7). Prior studies have reported swollen DVs in lines expressing the PfCRT variants C101F and L272F[27,55]. Notably, neither the PH1008-C nor the PH-1263C parents, nor the PH1008-C$^{Dd2\ crt}$ and PH1263-C$^{Dd2\ crt}$ edited lines, showed these unusual DVs. Unlike our *pfcrt*-variant Dd2 lines, these Cambodian parents harbored multicopy *pfpm2*, which suggests that multicopy *pfpm2* might ameliorate the distended DV phenotype associated with certain PfCRT mutations. The growth rates of these Cambodian isolates improved substantially following removal of M343L or H97Y, which coincided with *pfpm2* deamplification (Figs. 5, 6; Supplementary Tables 12, 13).

These data suggest a potential link between *pfpm2* copy number and variant PfCRT isoforms, which impacts PPQ susceptibility, DV morphology, and parasite growth rates. One connection could be the Hb degradation pathway. When compared with wild-type PfCRT, variant isoforms display elevated levels of partially degraded Hb peptides, thereby depriving the parasite of essential globin-derived amino acids[38,56]. This excess accumulation of peptides was earlier proposed to reduce parasite fitness, although more recent studies show a more complex relationship that likely involves other functional consequences of variant PfCRT. These include altered DV size, DV pH, and the intracellular amounts and ratios of Hb, free hematin, and hemozoin[38,56]. PfPM2 is a hemoglobinase active in the acidic DV[57,58]. Multicopy *pfpm2* might help overcome the fitness impact of PfCRT variants by increasing the rate of Hb degradation, perhaps leading to faster sequestration of reactive heme into chemically inert hemozoin. Metabolomics will be required to dissect the relationship between *pfpm2* copy number, *pfcrt* variants, and Hb digestion. Recent data have shown that deletion of *pfpm2* in the single-copy PPQ-sensitive 3D7 line caused a 1.5- or lower-fold decrease in PPQ IC$_{50}$ values compared to parental 3D7[59]. In that study, decreased parasite survival was observed using a modified PSA assay beginning with 0–8 h post-invasion (hpi) rings and very low concentrations of PPQ (6–12 nM). These concentrations are below the PSA$_{0–3h}$ concentration of 200 nM, applied against 0–3 hpi rings, which is associated with an increased risk of clinical failure[9,11]. Another recent study correlated multicopy *pfpm2* with the proliferation of PPQ-resistant parasites at elevated PPQ concentrations, as quantified using an area under the curve analysis[29]. Our efforts to genetically reduce the *pfpm2* copy number in Cambodian PPQ-resistant parasites have so far not yielded edited clones, despite PCR-based evidence of gene deletion events in bulk cultures, perhaps because of poor growth. Additional studies are needed to define the contribution of *pfpm2* gene amplification to PPQ resistance.

Our study highlights the pleiotropic effects of PfCRT mutations on multiple antimalarials whose modes of action intersect with Hb degradation. Each of the four amino acid substitutions studied herein impacted heme detoxification inhibitors (CQ, md-CQ, md-ADQ) in a mutation- and parasite line-dependent manner. These results highlight the major role of PfCRT in dictating *P. falciparum* susceptibility to heme-targeting drugs and the importance of the genetic background in modulating these effects[4]. These variants also impacted parasite susceptibility to LMF and MFQ, which are thought to accumulate in the DV away from their cytosolic site of action, as well as QN, whose complex mode of action includes inhibition of heme detoxification[4]. These PfCRT variants also showed minor effects on the ART metabolite

DHA, whose activation in mid-rings to trophozoites is thought to result primarily from interaction with $Fe^{2+}$-heme[60,61]. Dissecting these patterns is useful in predicting which drugs could best be deployed to treat multidrug-resistant malaria. This strategy is exemplified by recent clinical trials (NCT02612545 and NCT02453308) of the triple-combination therapies artemether + LMF + ADQ or DHA + PPQ + MFQ, as for these partner drugs the predominant PfCRT and PfMDR1 haplotypes exert opposing selective pressures[62].

The recent and rapid enrichment in Cambodian parasites of PfCRT mutations that mediate high-grade PPQ resistance following the adoption of DHA + PPQ highlights the need for full-length *pfcrt* sequencing in resistance surveillance studies. Obtaining these results quickly will be an important component of current efforts to track the spread of multidrug-resistant *P. falciparum* malaria and adjust treatment policy accordingly. Given the central role of heme detoxification in the mode of action of multiple antimalarials, the discovery of an unexpectedly diverse pool of *pfcrt* variants in Cambodia underscores the risk that drug-pressured selective sweeps could rapidly compromise new treatments.

## Methods

**Genome analysis of Pf3K field isolates.** This was performed on data generated by the Pf3K project (https://www.malariagen.net/projects/pf3k; pilot data release 3) comprising 2512 samples from 14 countries[23,37]. Assembly files were downloaded from the publicly accessible database (ftp://ngs.sanger.ac.uk/production/pf3k/release_3/BAM), and SNPs were manually extracted using the Pf3D7 reference genome version 11.0[63]. To this we added an additional 87 genomes obtained from Cambodian isolates in 2012 and 2013, deposited at the European Bioinformatics Institute[37]. We first identified all SNPs and rare variants present in the *pfcrt* gene of the field isolates based on the criteria of alternate allele frequency > 0.4 and the number of alternate reads ≥ 5. To confidently determine the PfCRT haplotypes, we filtered out samples where the alternate allele count at 12 codon positions (74, 75, 76, 144, 148, 194, 220, 271, 326, 333, 356, and 371) was <5. This resulted in 869 Asian genomes (Supplementary Table 1). Genome sequence data from the Pasteur Institute was generated from Cambodian parasites obtained from *P. falciparum*-infected patients who provided samples under a protocol (099 NEHR) approved on March 15, 2016 by the Cambodian National Ethics Committee for Health Research of the Ministry of Health.

**Plasmid construction.** The pZFN[14/15]-*bsd* and pcrt[Dd2]-h*dhfr* plasmids have been previously reported[40]. pZFN[14/15]-*bsd* expresses a pair of ZFNs specific to the intron 1/exon 2 junction of *pfcrt* (http://plasmodb.org/plasmo/app/record/gene/PF3D7_0709000). ZFNs were linked via a viral 2 A ribosome skip peptide. The DNA repair template was provided on the pcrt[Dd2+SNP]-h*dhfr* plasmid that carries exon 1, intron 1, and exons 2 to 13 of Dd2 *pfcrt*. This plasmid also harbors the SNP for each described mutation (H97Y, F145I, M343L, or G353V), which was introduced via site-directed mutagenesis into pcrt[Dd2]-h*dhfr* using primers p9–p16 (Supplementary Table 14). This *pfcrt* sequence is flanked by a *pfcrt* 5' untranslated region (UTR) and a *Plasmodium berghei crt* 3' UTR. Homology-directed repair (using *pfcrt* 5'- and 3'-UTR sequences as 5' and 3' regions of homology) resulted in integration of this modified *pfcrt* sequence, along with a downstream human *dhfr* marker that mediates resistance to the antiplasmodial agent WR99210 (Jacobus Pharmaceuticals).

**Parasite culturing and transfections.** Dd2 and the Cambodian isolates (PH1008-C and PH1263-C) were kindly provided by Drs. Thomas Wellems and Rick Fairhurst (Laboratory of Malaria and Vector Research, NIAID, NIH), respectively. *P. falciparum* ABS parasites were cultured in human O⁺ red blood cells (RBCs) in RPMI 1640-based culture media containing 0.5% (w/v) AlbuMAXII[64]. Parasite cultures were maintained at 3% hematocrit at 37 °C in an airtight chamber filled with 5% $O_2$/5% $CO_2$/90% $N_2$. Transfections were performed by electroporating ring-stage parasites at 5–10% parasitemia with 50 μg of purified circular plasmid DNA in Cytomix[64]. Parasites were first transfected with the donor plasmid pcrt[Dd2 crt]-h*dhfr* and selected with 2.5 nM WR99210 to enrich for episomally transformed parasites. These parasites were then further transfected with pZFN[14/15]-*bsd* and selected in 2 μg/mL blasticidin (ThermoFisher) for 6 days. Parasites were visible microscopically 3 to 6 weeks post-electroporation and were screened for editing via PCR. Positively edited bulk cultures were cloned via limiting dilution in 96-well plates containing an average of 0.3 parasites per well. These plates were screened for viable parasites after 17 days via flow cytometry. Briefly, cells were stained with 100 nM MitoTracker Deep Red and 1× SYBR green (ThermoFisher) in 1× phosphate-buffered saline (PBS pH 7.4), incubated at 37 °C for at least 20 min in the dark, and quantified on an Accuri C6 flow cytometer (Becton Dickinson). Approximately 10,000 events were read per well. Positively edited clones were expanded for DNA and phenotypic analysis. Of note, the line from which Dd2 was cloned was first adapted to culture in

the early 1980s[65], whereas the Cambodian isolates[37] were adapted in 2012–13, after the adoption of DHA + PPQ as first-line therapy in 2008.

**DNA analysis of clones.** *pfcrt* editing events were confirmed using a PCR-based approach (Supplementary Fig. 3; Supplementary Table 14). PCR amplification was performed on genomic DNA using primer pairs p1 + p2 (edited, 2.5 kb; unedited, no product), p3 + p4 (edited, 0.4 kb; unedited, 0.6 kb), and p5 + p6 (edited 2.0 kb; unedited, 3.5 kb). Removal of all introns (except for the first) from the edited Dd2 parasites resulted in shorter PCR amplicons compared to the unedited *pfcrt* gene. The presence of the desired sequence in these edited lines was confirmed by Sanger sequencing using primers p3, p6, and p7.

**Piperaquine survival assay.** PSAs were performed with 0–3 hpi ring-stage parasite cultures[11]. To determine PPQ survival, we isolated 0–3 hpi rings following 70% Percoll gradients (GE Healthcare) or MACS LD columns (Miltenyi Biotec). We incubated these tightly synchronized ring parasites at 0.5% starting parasitemia and 1% hematocrit with 200 nM PPQ tetraphosphate tetrahydrate (Alfa Aesar) or 0.5% lactic acid in water (vehicle control) at 37 °C for 48 h in 96-well plates (Sigma). The cultures were then washed three times in separate U-bottom plates and re-plated in fresh 96-well flat-bottom plates. Cultures were continued for another 24 h without drug treatment. Parasitemias were determined via flow cytometry, as described above for in vitro $IC_{50}$ drug susceptibility assays. 30,000–50,000 events were read per well. Initial testing found good concordance between flow cytometry data collection and that from the more labor-intensive microscopy-based counting from Giemsa-stained thin blood smears. Statistical comparisons between cell lines were made using Mann–Whitney $U$ tests. Assays were repeated in duplicate at least twice.

Susceptibility to PPQ was defined as the survival rate calculated using the formula listed below. A survival rate >10% was deemed resistant.

$$\text{PSA survival rate} = 100 \times \frac{\text{Number of viable parasites in exposed culture}}{\text{Number of viable parasites in vehicle culture}} \quad (1)$$

**In vitro $IC_{50}$ drug susceptibility assays.** We tested for changes in the in vitro susceptibility of the genetically edited parasite clones to different antimalarials by comparing their $IC_{50}$ values to control cell lines. $IC_{50}$ values were determined for PPQ, CQ, md-CQ, md-ADQ, QN, MFQ, LMF, and DHA (Supplementary Fig. 1). md-CQ and md-ADQ are the in vivo metabolites of CQ and ADQ, respectively. To determine $IC_{50}$ values, we incubated parasites at 37 °C, with 0.2% starting parasitemia and 1% hematocrit across a range of drug concentrations with two-fold dilutions in 96-well plates. Parasite growth in each well was assessed after 72 h using flow cytometry, as described above. Approximately 10,000 events were read per well. For PPQ, $IC_{50}$ and $IC_{90}$ values were extrapolated by linear regression, because of the unusual dose–response curves observed in resistant lines (Fig. 3). For all other antimalarials, the in vitro $IC_{50}$ values were determined by nonlinear regression analysis with GraphPad Prism 7 software. Statistical comparisons between cell lines were made using Mann–Whitney $U$ tests. Assays were repeated in duplicate on 4–15 independent occasions.

**Piperaquine accumulation assays.** Accumulation of [³H]-PPQ (15 Ci/mmol; American Radiolabeled Chemicals) was measured with synchronized trophozoites[66]. Briefly, 250 μL of parasites (at 2% hematocrit, ~5% parasitemia) or uninfected red cells (2% hematocrit) were washed with bicarbonate-free RPMI media (supplemented with 25 mM HEPES, 10 mM glucose, and 0.2 mM hypoxanthine, adjusted to pH 7.4) and added to an equal volume of 20 nM [³H]-PPQ in bicarbonate-free media in a 1.5 mL Eppendorf tube. After incubation at 37 °C for 1 h, 200 μL aliquots were transferred to tubes containing 400 μL of dibutyl phthalate (Sigma Aldrich, 1.04 g/mL) and centrifuged immediately (20,800×*g*, 2 min) to sediment the cells through the oil. Radioactivity in supernatants and solubilized cell pellets from infected and uninfected control RBCs was then measured[66]. PPQ CARs were calculated using an estimate of the volume of a trophozoite-infected erythrocyte (75 fL)[67]. PPQ accumulation was expressed as the ratio of the intracellular versus extracellular PPQ concentration. Data were collected from three to seven independent experiments with duplicate technical repeats.

**Fitness assays.** We evaluated the impact of PfCRT mutations on in vitro growth rates over 10 generations, as a proxy for fitness[68]. To do so, we seeded 1:1 co-cultures of each line of interest with Dd2attB-eGFP, a line that constitutively expresses a single copy of *eGFP* under the *calmodulin* promoter. We then propagated the co-cultures for 20 days and quantified the percentage of eGFP⁺ cells by flow cytometry three times per week. Flow cytometry was performed as above, but only with 100 nM MitoTracker Deep Red staining. SYBR Green fluoresces in the same channel as eGFP and thus was not used. Approximately 10,000 events were read per well. Assays were repeated in triplicate two times.

**Quantitative PCR-based determination of *pfpm2* copy number.** Genomic DNA was extracted using 0.1% saponin lysis followed by the QIAamp Blood Kit (Qiagen)

with RNase treatment. *pfpm2* (http://plasmodb.org/plasmo/app/record/gene/PF3D7_1408000) copy number was determined by multiplexed qPCR of the *pfpm2* and the single copy Pf *β-tubulin* (http://plasmodb.org/plasmo/app/record/gene/PF3D7_1008700) reference control gene in Taqman assays on a StepOne Plus (ThermoFisher Scientific). For every qPCR run, a standard curve consisting of ten-fold serially diluted DNA template was generated to measure reaction efficiency, and five standards of mixed gene fragments at 1:1, 2:1, 3:1, 4:1, and 5:1 molar ratio of *pfpm2* & *β-tubulin* were included as copy number controls. qPCR reactions consisted of 12 ng of genomic DNA template, 0.16 μM final concentration of each TaqMan probe, and 0.33 μM final concentration of each set of forward and reverse primers using the SsoAdvanced™ Universal Probes Supermix (BioRad) for a total reaction volume of 20 μL. Four independent reactions of each sample in technical triplicates was carried out with the following conditions: 3 min at 95 °C, 20 s at 95 °C, 40 s at 58 °C for 45 cycles. Copy number was determined using the standard curve method for relative quantification.

$$\text{RQ} = \left(E_{pfpm2}\right)^{\Delta Ct(pfpm2)} / \left(E_{\beta-tubulin}\right)^{\Delta Ct(\beta-tubulin)} \qquad (2)$$

where $\Delta Ct(pfpm2)$ = Ct(*pfpm2* in 3D7) – Ct(*pfpm2* in sample) and $\Delta Ct(\beta\text{-}tubulin)$ = Ct(*β-tubulin* in 3D7) – Ct(*β-tubulin* in sample) and $E$ is the primer efficiency derived from the standard curves.

**Reverse transcription quantitative PCR**. Total RNA was extracted from highly synchronized trophozoites for each parasite line using the Direct-zol™ RNA MiniPrep kit (Zymo Research) and was DNaseI-treated to remove any contaminating genomic DNA. *pfpm2* transcript levels were determined by multiplexed RT-qPCR of the *pfpm2* and *serine tRNA ligase* (http://plasmodb.org/plasmo/app/record/gene/PF3D7_0717700) reference gene. For every run, five positive controls of 10-fold serially diluted DNAseI-treated RNA (20, 2, 0.2, 0.02, 0.002 ng) were included. RT-qPCR reactions comprised of 12 ng of total RNA, 0.16 μM final concentration of each TaqMan probe, 0.33 μM final concentration of each set of forward and reverse primers, and 16 units of RNase-OUT and SuperScript™ III Platinum™ One-Step qRT-PCR Kit (Invitrogen) for a total reaction volume of 20 μL. Three independent reactions of each sample were performed in technical triplicates, with the following cycling conditions: 15 min at 50 °C (reverse transcription); 2 min at 95 °C; then 45 cycles of 20 s at 95 °C and 40 s at 58 °C. Control reactions lacking Superscript Reverse transcriptase enzyme yielded no Ct value, confirming the absence of contaminating genomic DNA. The relative expression of *pfpm2* was determined using the following formula:

$$\text{RQ} = \left(E_{pfpm2}\right)^{\Delta Ct(pfpm2)} / \left(E_{tRNA\_ligase}\right)^{\Delta Ct(tRNA\_ligase)} \qquad (3)$$

where $\Delta Ct(pfpm2)$ = Ct(*pfpm2* in 3D7) – Ct(*pfpm2* in sample) and $\Delta Ct(tRNA\_ligase)$ = Ct(*tRNA_ligase* in 3D7) – Ct(*tRNA_ligase* in sample), where $E$ is the primer efficiency derived from the standard curves. Primers and probes are listed in Supplementary Table 14.

**Data availability**. The authors declare that the data supporting the findings of this study are available within the paper and its supplementary information, or are available from the authors upon request. Genome data from the Pf3K project are available at https://www.malariagen.net/projects/pf3k. Assembly files are publicly accessible at ftp://ngs.sanger.ac.uk/production/pf3k/release_3/BAM.

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

## Acknowledgements

Funding for this work was provided in part by R01 AI50234 and R01 AI124678 (to D.A. F.), NRSA fellowship F32 AI120578 (to L.S.R.), R01 AI125579 (to S.T.H.), and the Intramural Research Program of the NIAID, NIH (R.M.F.). The work of S.M. was supported by the Human Frontier Science Program (HFSP) postdoctoral fellowship. We thank Rithea Leang and Melissa Mairet-Khedim (Pasteur Institute in Cambodia) for their help with isolate collection and processing, and Philipp P. Henrich (Columbia University) for his initial help with analyzing the Pf3K database.

## Author contributions

Conceived and designed the experiments: L.S.R., S.K.D., D.A.F. Acquired the data: L.S.R., S.K.D., S.M., K.J.W., K.K., S.T.H., B.W., R.M.F., F.A., D.M. Analyzed and interpreted the data: L.S.R., S.K.D., S.M., T.Y., K.J.W., F.A., D.M., D.A.F. Wrote the paper: L.S.R., D.A.F., with input from other authors. All authors approved the final manuscript.

## Additional information

**Competing interests:** The authors declare no competing interests.

