## [Peer Review File · Nature Communications]

Reviewers' comments:

Reviewer #1 (Remarks to the Author):

Ross et al make a compelling case for the previously proposed importance of pfCRT as an emerging factor in the development of piperazine resistance.

The starting step for these investigations were observations of the rise of new mutations in the pfCRT gene in several regions using piperazine, supporting a previous (to a certain extent unclear) publication of selection of resistance in vitro. This is an interesting and well conducted study. The report is well written.

Some comments are anyway due.

The authors should provide functional data supporting an associated resistance mechanism, albeit briefly discussing it in the discussion. Basically, is pfCRT a piperazine transporter as it has been found concerning CQ? An approach similar to the employed by Martin et al (Martin, Science, 2009) would likely give important insights.

Also, and somewhat related, it was surprising that the effects of verapamil were not tested. Revisiting the pfCRT edited clones for understanding if the PSA (0-3) survival rates are significantly changed would be of interest, taking in consideration the importance of the Dd2 (76T carrying) background. Such a biochemical tool could help understanding the interesting sensitization effect of these new PPQ-R SNPs on CQ, and potentially the relation between them and pfPM2 increased copy number.

Finally, I think that the authors should be more careful on the interpretation of the pfmdr1 duplications data. It seems evident (and actually referred by the authors) that these mutations are only present among sensitive infections. If on one side one can argue that the heterogeneity of this group (from 1-4 copies) points for a non significant role of this mutation, on the other hand it seems that it is not compatible with PPQ resistance. One alternative explanation is that pfmdr1 duplications give PPQ hypersensitivity to the parasite, probably depending on the genetic background. This can be explained by this transporter concentrating PPQ in this Food Vacuole (FV) lumen, as it is believed to be a drug importer and/or due to fitness costs that are incompatible with the PPQ resistance phenotype. Associations between pfmdr1 duplications and PPQ increased sensitivity have been observed before (albeit using conventional IC50 measurements)(Veiga, AAC, 2013), while a clear decrease in pfmdr1 duplications has been witnessed upon the implementation of DHA-PPQ in Cambodia (Imwong, AAC, 2010).

Reviewer #2 (Remarks to the Author):

Having reviewed the manuscript "Emerging Southeast Asian PfCRT mutations confer Plasmodium falciparum resistance to the first-line antimalarial piperazine", my recommendation is to publish this great work basically as it is. This important work is well and concisely presented !

The analysis of 869 Asian *P. falciparum* genomes sequenced by the Pf3K consortium and of an additional 93 Cambodian genomes collected by investigators at the Pasteur Institute in Cambodia suggests a rapid increase in the prevalence of novel variants, and provides evidence of a remarkable array of novel PfCRT mutations. These findings illustrate the need to survey the emergence of novel PfCRT mutations in field isolates to ascertain their association with PPQ treatment failures in patient populations.

Minor, but still essential revisions before publication

- Abstract: For further clarity, please add somewhere the information that the found PfCRT mutations do arise on a mutated K13 background, like it is also mentioned in the last paragraph on page 5.

- Page 15 and 16: The words "of note" is used at least three times.

- Page 17: DHA is the principal metabolite of all artemisinin derivatives, but not artemisinin per se, please correct (as a reference, see "Clin Pharmacokinet. 2000 Oct;39(4):255-70. Pharmacokinetics of artemisinin-type compounds. Navaratnam V1, Mansor SM, Sit NW, Grace J, Li Q, Olliaro P.")

NCOMMS-18-07241, Ross *et al.*, Resubmission

Point-by-point reply to comments from the Editor and Reviewers. Below we indicate deleted text in red with a strikethrough, and new text in blue. Corresponding changes are highlighted in yellow in the revised manuscript. We have also added two new **Supplementary Tables 6 and 11** for the [³H]-piperazine accumulation and the verapamil data, respectively, with appropriate renumbering. We also moved former **Supplementary Figure S4** into the main text as **Figure 7**, in light of its importance to the study. We have also gone through the manuscript and made minor grammatical improvements, and revised the references to incorporate a new relevant publication and trim the number to 70.

Editor comments:

We are interested in the possibility of publishing your study in Nature Communications, but would like to consider your response to [comments raised by 2 referees] in the form of a revised manuscript before we make a final decision on publication...Please note that we do agree with reviewer #1 that additional experiments providing insights into mechanistic basis of the observations would strengthen the case for publication with us. Please make sure that a revised manuscript addresses these and all other concerns in full, and highlight all changes in the manuscript text file.

Reply: We would like to thank the editor for this positive assessment of our submission and giving us the opportunity to address concerns raised by the two reviewers. Our revised manuscript now contains additional data as requested by Reviewer 1, accompanied by further discussion of the mechanistic basis of our observations.

Reviewer #1 Comments:

Comment 1: *The authors should provide functional data supporting an associated resistance mechanism, albeit briefly discussing it in the discussion. Basically, is pfCRT a piperazine transporter as it has been found concerning CQ? An approach similar to the employed by Martin et al (Martin, Science, 2009) would likely give important insights.*

Reply: We thank the reviewer for this insightful question. In the 2009 report by Martin *et al.*, published in *Science*, chloroquine (CQ) transport studies were conducted with *Xenopus* oocytes that expressed PfCRT isoforms at their surface. Those studies provided compelling evidence that mutant PfCRT isoforms capable of mediating CQ resistance do so at least in part by acquiring CQ transport properties, consistent with the idea that, in *Plasmodium falciparum* asexual blood stage parasites, mutant PfCRT effluxes CQ out of the digestive vacuole and away from its heme ligand. The reviewer astutely asks whether a similar mechanism is at play here, namely is piperazine (PPQ) transport mediated by the novel PfCRT variants that we describe herein? To address this, we have performed [³H]-PPQ accumulation assays with our panel of parasites, including ones that differ in their *pfCRT* allele and PPQ susceptibility. Accumulation studies were performed on 3 to 7 independent occasions, with technical replicates. These data are presented in the new **Figure 3f** panel and detailed in the new **Supplementary Table 6**. The corresponding changes to the text are listed below.

Results (beginning page 11, 3rd paragraph): “Uptake of radiolabeled PPQ into parasitized erythrocytes is dependent on the parasite genetic background. To examine the role of PfCRT mutations on PPQ accumulation in our parental versus edited cell lines, we measured [³H]-

PPQ uptake into synchronized trophozoites. Cultures were incubated with 10 nM [³H]-PPQ at 37°C for 1 h and radioactivity was measured separately for cells and supernatants. From this, we derived the ratio of intracellular versus extracellular [³H]-PPQ concentration, representing the cellular accumulation ratio (CAR) (Fig. 3f; Supplementary Table 6). The most significant difference in the CAR was observed between cell lines expressing the CQ- and PPQ-sensitive 3D7 *pfcr*t allele and all others expressing the CQ-resistant Dd2 *pfcr*t allele or its PPQ-resistant variants. Compared to 3D7, the CAR was ~2-fold and ~2 to 4-fold less in the Dd2 and Cambodian variants, respectively. These data suggest that the wild-type 3D7 PfCRT isoform allows for increased PPQ accumulation in the DV, where retention is predicted to be driven by the weak-base gradient and PPQ interactions with hemozoin²⁸. Notably, there was no significant difference between the edited Dd2^{Dd2 crt} PPQ-sensitive control line versus the Dd2 PPQ-resistant variants expressing the PfCRT mutations F145I, M343L, or G353V, suggesting that these mutations do not confer resistance via altered PPQ accumulation in the Dd2 background. Interestingly, there was a ~1.8-fold increase in the CAR for the edited recombinant clones PH1008-C^{Dd2 crt} and PH1263-C^{Dd2 crt}, which were sensitized to PPQ, compared with their parental PPQ-resistant isolates. The influence of the genetic background on PPQ uptake in PPQ-resistant versus sensitive variants highlights the complex relationship between these PfCRT mutations and PPQ resistance, which cannot be explained solely by changes in drug accumulation or efflux.”

Discussion (beginning page 19, 3rd paragraph): “It is well established that PfCRT isoforms giving rise to CQ resistance, such as Dd2, are able to transport CQ away from their primary site of action via a DV efflux mechanism, reducing intracellular drug accumulation in CQ-resistant strains⁴⁹⁻⁵³. We examined whether the novel PfCRT isoforms might similarly mediate PPQ transport, given the structural similarity between PPQ and CQ and the evidence that both drugs accumulate to high concentrations in the DV and inhibit β-hematin formation⁵⁴. Indeed, our recent Hb fractionation studies found both drugs to be similar in their inhibition of hemozoin formation and the resulting buildup of the reactive heme precursor²⁸. We found no significant difference in [³H]-PPQ accumulation between our edited Dd2 lines expressing the F145I, M343L, G353V, or H97Y mutations compared to Dd2 *pfcr*t. With the Cambodian isolates, we observed only a relatively small increase in accumulation in the revertant PPQ-sensitive lines compared to the resistant parents. These results suggest that this variant PfCRT-mediated PPQ resistance mechanism is not primarily a function of changes in intracellular drug accumulation. One alternative explanation might involve PPQ-mediated binding to and functional inhibition of certain PfCRT isoforms, recalling earlier reports of distinct drug binding sites in this transporter^{52,55,56}. Further investigations in gene-edited parasites and existing *Xenopus* oocyte- or yeast-based heterologous expression systems^{53,56} will be important in delineating the relationship between PfCRT mutations, *pfpm2* copy number, drug accumulation, and PPQ resistance.”

Methods (beginning page 27, 4th paragraph): “**Piperaquine accumulation assays.** Accumulation of [³H]-PPQ (15 Ci/mmol; American Radiolabeled Chemicals) was measured essentially as reported⁶⁹. Briefly, 250 μL of synchronized trophozoites (at 2% hematocrit, ~5% parasitemia) or uninfected red cells (2% hematocrit) were washed with bicarbonate-free RPMI media (supplemented with 25 mM HEPES, 10 mM glucose and 0.2 mM hypoxanthine, adjusted to pH 7.4) and added to an equal volume of 20 nM [³H]-PPQ in bicarbonate-free media in a 1.5 mL Eppendorf tube. After incubation at 37°C for 1 h, 200 μL aliquots were transferred to tubes containing 400 μL of dibutyl phthalate (Sigma Aldrich, 1.04 g/mL) and centrifuged immediately

(14000 rpm, 2 min) to sediment the cells through the oil, terminating the reaction. Radioactivity in supernatants and solubilized cell pellets from infected and uninfected control RBCs was measured as described⁶⁹. PPQ cellular accumulation ratios were calculated using an estimate of the volume of a trophozoite-infected erythrocyte (75 fL)⁷⁰. PPQ accumulation was expressed as the ratio of the intracellular versus extracellular PPQ concentration. Data were collected from 3 to 7 independent experiments with duplicate technical repeats.”

Figure 3 Legend (beginning page 38, 2nd paragraph): “(f) Intracellular versus extracellular [³H]-PPQ concentration after 1h incubation at 37°C with synchronized trophozoites (when the DV is fully formed and hemoglobin degradation is maximal). Results show the cellular accumulation ratios (CAR) for each parasite line, presented as means ± SEMs (N,n = 3 to 7, 2; **Supplementary Table 6**). Significance was determined using Mann-Whitney *U* tests comparing *pfCRT*-edited parasites and their isogenic controls, as well as Dd2^{Dd2 crt} versus 3D7. **p*<0.05; ***p*<0.005; ****p*<0.0005; *****p*<0.0001.”

Comment 2: *Also, and somewhat related, it was surprising that the effects of verapamil were not tested. Revisiting the pfCRT edited clones for understanding if the PSA (0-3) survival rates are significantly changed would be of interest, taking in consideration the importance of the Dd2 (76T carrying) background. Such a biochemical tool could help understanding the interesting sensitization effect of these new PPQ-R SNPs on CQ, and potentially the relation between them and pfPM2 increased copy number.*

Reply: We have now carried out CQ and PPQ IC₅₀ assays and PSA_{0-3h} assays in the presence or absence of 800 nM verapamil (VP). Results showed that VP strongly sensitized Dd2 parasites to CQ, corresponding to a ~72% decrease in IC₅₀ values for CQ + VP when compared to CQ alone. No sensitization was observed with the CQ-sensitive lines 3D7 and GC03. These degrees of reversibility very closely match our earlier published observations. The PPQ-resistant Dd2^{Dd2 crt F145I}, Dd2^{Dd2 crt M343L}, and Dd2^{Dd2 crt G353V} parasite lines showed were both less CQ-resistant and less VP reversible. Our results with PPQ were quite different in that these novel polymorphisms did not substantially alter levels of PPQ resistance. Results were complicated by two features of PPQ resistance: flat dose-response curves, as documented in multiple studies, and evidence of increased toxicity of VP alone in parasites harboring these novel PfCRT mutations. These assays will therefore require substantial optimization to define the dose-dependent effect of VP toxicity itself and the effect of VP on PPQ susceptibility across different strains and assays. Nonetheless, our PSA_{0-3h} assays, which have now been repeated several times with incorporation of these new data, revealed no reversibility with the Dd2^{Dd2 crt F145I} line. This line showed the highest degree of resistance both by PSA and in our dose-response assays. These data provide evidence that the PPQ resistance phenotype is not VP reversible. This difference from PfCRT-mediated CQ resistance is consistent with our [³H]-PPQ data that do not show the reduced level of drug accumulation that many groups have generally observed with mutant PfCRT-mediated CQ resistance. Our VP results are now summarized in **Supplementary Table 11** and are incorporated into our manuscript as follows:

Results (beginning page 14, 2nd paragraph): “We also explored the effects of verapamil (VP), a CQ resistance reversal agent, on CQ and PPQ susceptibility⁴⁵. For Dd2 and Dd2^{Dd2 crt}, we observed a ~72% reduction in CQ IC₅₀ values in the presence of 0.8 μM VP, consistent with earlier reports^{42,46}. The addition of the F145I, M343L, or G353V mutations led to less reversibility (26–53%), differing significantly from Dd2^{Dd2 crt} (**Supplementary Table 11**). No

reversibility was observed with the CQ-sensitive 3D7 and GC03 lines. In the parental and edited Cambodian lines, VP reversibility was observed at 60–65%, irrespective of their PfCRT haplotype or *pfpm2* copy number. With PPQ, we observed 17–43% reversibility among the PPQ resistance-conferring variants on the Dd2 background; however, these differences were not significant compared to the Dd2^{Dd2 crt} isogenic control. PPQ-sensitive lines displayed minimal VP reversibility (2–17%). Two factors complicated our analysis of the effect of VP on PPQ IC₅₀ values. First, 0.8 μM VP on its own inhibited growth by 18–45% in the lines expressing a novel PfCRT mutation (H97Y, F145I, M343L or G353V), compared to <14% inhibition for all other lines (**Supplementary Table 11**). Second, dose-response profiles were quite flat around the region of 50% inhibition, creating sizable variation (**Figure 3**).

PSA_{0-3h} tests also revealed no significant impact of 0.8 μM VP on percent survival of either Dd2^{Dd2 crt F145I}, which had the highest level of PPQ survival, or the PPQ-sensitive lines Dd2^{Dd2 crt}, Dd2, 3D7, or GC03 (**Supplementary Table 4**). Taken together, these results provide evidence that VP does not substantially reverse PPQ resistance, unlike with CQ, highlighting mechanistic distinctions between PfCRT-mediated CQ and PPQ resistance.”

Discussion (page 20, 2nd paragraph): “Additional evidence that PfCRT-mediated CQ and PPQ resistance differ mechanistically was obtained by analyzing the effect of VP, which strongly reversed CQ but not PPQ resistance. In the Dd2 background, the novel PfCRT mutations significantly reduced the degree of VP reversibility on CQ and also resulted in greater toxicity of VP alone, when compared with parasites expressing the Dd2 isoform. Unexpectedly, all four Cambodian lines displayed similar levels of VP reversibility for CQ, despite differences in their PfCRT haplotype and *pfpm2* copy number. This finding suggests that other loci in the Cambodian isolates might modify the VP response, which until now has only been linked to PfCRT⁵⁷.”

Comment 3: Finally, I think that the authors should be more careful on the interpretation of the *pfmdr1* duplication data. It seems evident (and actually referred by the authors) that these mutations are only present among sensitive infections. If on one side one can argue that the heterogeneity of this group (from 1-4 copies) points for a non-significant role of this mutation, on the other hand it seems that it is not compatible with PPQ resistance. One alternative explanation is that *pfmdr1* duplications give PPQ hypersensitivity to the parasite, probably depending on the genetic background. This can be explained by this transporter concentrating PPQ in this Food Vacuole (FV) lumen, as it is believed to be a drug importer and/or due to fitness costs that are incompatible with the PPQ resistance phenotype. Associations between *pfmdr1* duplications and PPQ increased sensitivity have been observed before (albeit using conventional IC₅₀ measurements) (Veiga, AAC, 2013), while a clear decrease in *pfmdr1* duplications has been witnessed upon the implementation of DHA-PPQ in Cambodia (Imwong, AAC, 2010).

Reply: We agree with the reviewer that prudence is required when assessing the role of *pfmdr1* copy number in PPQ resistance. In a prior publication from our lab, we showed that isogenic FCB parasites expressing one or two copies of *pfmdr1* showed no difference in their mean PPQ IC₅₀ values. Nonetheless, contemporary Cambodian PPQ-resistant parasites (as shown using the PSA) were found in our study to have only 1 or 2 *pfmdr1* copies, whereas ones with 3 or 4 copies were sensitive. We have now amended our text to incorporate the references mentioned above and to qualify our interpretation, as follows:

Introduction (page 6, 2nd paragraph), we replaced the sentence: “~~Increased copy number of *pfmdr1*, which like *pfprt* encodes a DV membrane-resident transporter and is a common culprit for drug resistance, was found to be unrelated to PPQ IC₅₀ values in clinical isolates⁴⁰ and in *pfmdr1*-edited parasite lines³⁸.~~” with the following text: “*pfmdr1*, which like *pfprt* encodes a DV membrane-resident transporter, has also been considered. Studies have revealed a reduced prevalence of multicopy *pfmdr1* since the implementation of DHA+PPQ^{8,10,31,32}, and an earlier report documented a mild inverse association between *pfmdr1* copy number and PPQ IC₅₀ values³³. Other studies, including with isogenic parasites differing only in *pfmdr1* copy number, found no direct association with PPQ IC₅₀ values^{28,34}. In the field, *pfmdr1* deamplification might also result from less use of MFQ, an ACT partner drug that is compromised by *pfmdr1* amplification. Reduced fitness of multicopy *pfmdr1* favors deamplification without MFQ pressure^{35,36}.”

Results (page 9, 1st paragraph), we also replaced “~~PPQ sensitive parasites were found to have 1–4 copies of *pfmdr1*, consistent with an earlier report using recombinant isogenic lines that showed no direct association between *pfmdr1* copy number and levels of PPQ susceptibility³⁸.~~” with “All PPQ-resistant parasites were single copy for *pfmdr1*, in comparison to PPQ-sensitive parasites that had 1–4 *pfmdr1* copies.”

Reviewer #2 Comments:

Having reviewed the manuscript “Emerging Southeast Asian PfCRT mutations confer Plasmodium falciparum resistance to the first-line antimalarial piperaquine”, my recommendation is to publish this great work basically as it is. This important work is well and concisely presented! The analysis of 869 Asian P. falciparum genomes sequenced by the Pf3K consortium and of an additional 93 Cambodian genomes collected by investigators at the Pasteur Institute in Cambodia suggests a rapid increase in the prevalence of novel variants, and provides evidence of a remarkable array of novel PfCRT mutations. These findings illustrate the need to survey the emergence of novel PfCRT mutations in field isolates to ascertain their association with PPQ treatment failures in patient populations.

Minor, but still essential revisions before publication

- Abstract: For further clarity, please add somewhere the information that the found PfCRT mutations do arise on a mutated K13 background, like it is also mentioned in the last paragraph on page 5.

Reply: We thank the reviewer for raising this question, which we have interrogated using our panel of 93 isolates sequenced by our colleagues at the Pasteur Institute in Cambodia. This yielded the interesting observation that all of the novel PfCRT variants that had evolved on the Dd2 haplotype were present in parasites with the K13 C580Y mutation. In contrast, isolates with the Dd2 haplotype had a mixture of K13 mutant and wild-type alleles. Corresponding changes are listed below:

Abstract (page 2), we now state: “These mutations occurred in parasites harboring the K13 C580Y artemisinin resistance marker.”

Results (page 8, 2nd paragraph), we now include: “Intriguingly, of the 34 isolates harboring a novel PfCRT mutation, all were on the Dd2 PfCRT background and all carried the K13 C580Y

mutation. In contrast, parasites expressing the PfCRT Dd2 haplotype without additional mutations were a mixture of mutant and wild-type K13, with C580Y and R539T being predominant. Parasites expressing the less prevalent GB4 and Cam783 PfCRT haplotypes were all K13 wild-type (**Supplementary Table 3**).”

- Page 15 and 16: The words “of note” is used at least three times.

Reply: We thank the reviewer for pointing out this over-saturation of the term. Of note has now been removed from the 2nd and 3rd instances of its earlier use (page 19).

- Page 17: DHA is the principal metabolite of all artemisinin derivatives, but not artemisinin per se, please correct (as a reference, see “Clin Pharmacokinet. 2000 Oct;39(4):255-70. Pharmacokinetics of artemisinin-type compounds. Navaratnam V1, Mansor SM, Sit NW, Grace J, Li Q, Olliaro P.”)

Reply: This change has been made (page 22, 2nd paragraph), along with addition of this reference (now reference 63).

Comment 1: Page 17. Discussion of the mode-of-action results for PPQ should mention PMID: 17466277.

Reply: We have now incorporated this reference to earlier work into the following revised sentence in the **Results** (page 19, 3rd paragraph): “We examined whether the novel PfCRT isoforms might similarly mediate PPQ transport, given the structural similarity between PPQ and CQ and the evidence that both drugs accumulate to high concentrations in the DV and inhibit β -hematin formation⁵⁴.” This sentence is included in a revised paragraph described in our reply above to **Reviewer 1 Comment 1**.

REVIEWERS' COMMENTS:

Reviewer #1 (Remarks to the Author):

I appreciate the efforts of the authors on developing a good manuscript into an excellent one.

I consider that all my issues were addressed.

NCOMMS-18-07241A, Ross *et al.*, response to review of Resubmitted manuscript.

Our original submission had two reviews, with the first being detailed and the second having no substantive requests for changes. Our resubmission was reviewed again by Reviewer 1. The reviewer comments and our reply are listed below.

Reviewer #1 (Remarks to the Author):

I appreciate the efforts of the authors on developing a good manuscript into an excellent one. I consider that all my issues were addressed.

Reply: We thank the Reviewer for this positive assessment of our revised manuscript.